# MathViz-Bench: Evaluating Text-to-Image Models on Visually Solving Math Problems

## Abstract

We present MathViz-Bench, a comprehensive benchmark for evaluating Text-to-Image (T2I) models' capability to visualize step-by-step solutions for high school mathematics problems. MathViz-Bench comprises 500 carefully curated problems sampled from levels 1-3 of the MATH dataset, spanning seven mathematical domains: Prealgebra, Algebra, Number Theory, Counting & Probability, Geometry, Intermediate Algebra, and Precalculus. We transform these problems into prompts requiring models to generate visual step-by-step solutions with proper mathematical notation and logical flow. Our automated assessment pipeline employs three metrics: Sequential Consistency for logical flow, Symbol Fidelity for notation accuracy, and Mathematical Correctness for calculation validity, each scored 0-5. Our evaluation shows that models with built-in language understanding (GPT-Image-1: 84.05%, Gemini-2.5-Pro: 75.13%) perform much better than diffusion models (FLUX1.1-Pro: 35.23%, WAN2.2: 28.05%, Stable Diffusion 3.5 Ultra: 22.05%), achieving 2-3 times higher scores. All models exhibit high Symbol Fidelity (1.81-4.57) but fail at Mathematical Correctness (0.65-4.05), indicating they process mathematical symbols as visual patterns rather than semantic operators. Diffusion models demonstrate complete difficulty invariance and 33.8% critical failure rates, confirming absence of mathematical reasoning. These findings establish that mathematical visualization requires architectural integration of symbolic reasoning with visual generation, beyond current T2I capabilities.

## 1 Introduction

Text-to-Image (T2I) generation has emerged as one of the most successful applications of generative AI, with models like DALL-E 3 (OpenAI, 2023), Stable Diffusion 3 (Esser et al., 2024), Midjourney v6 (Midjourney Inc., 2023), and FLUX.1 (Black Forest Labs, 2024) producing remarkably photorealistic and creative visual content from natural language descriptions. These models have demonstrated impressive capabilities across diverse domains, from artistic creation and product design to scientific visualization and architectural rendering (Rombach et al., 2022; Saharia et al., 2022). Recent advances in diffusion models (Liu et al., 2022; Peebles & Xie, 2023) and vision-language pretraining (Radford et al., 2021; Li et al., 2023) have pushed the boundaries of what is possible, enabling fine-grained control over style, composition, and semantic content (Zhang et al., 2023; Mou et al., 2024). The widespread bosheah2025challenges of these tools by creative professionals, educators, and researchers underscores their transformative potential (Liu et al., 2024). However, this remarkable success masks a critical limitation: while T2I models excel at generating natural scenes and artistic content where minor inaccuracies are acceptable or even desirable, they fail when tasked with generating structured mathematical content that demands precise symbolic accuracy and logical reasoning (Kajić et al., 2024; Bosheah & Bilicki, 2025). Recent evaluations reveal systematic failures in mathematical symbol rendering (Bosheah & Bilicki, 2025), with state-of-the-art models struggling with basic numerical reasoning (Kajić et al., 2024), spatial consistency (Chatterjee et al., 2024), and the generation of pedagogically meaningful visualizations for mathematical content (Wang et al., 2025), rendering them unsuitable for educational applications where accuracy is paramount (Ji et al., 2025). Figure 1 illustrates these limitations. While GPT produces readable mathematical notation, specialized diffusion models generate illegible or nonsensical outputs, highlighting the fundamental disconnect between visual generation and mathematical reasoning in current architectures. More detailed examples can be found at Appendix A.

Despite mathematics education's dependence on visual demonstrations, no existing benchmark evaluates T2I models' ability to generate accurate mathematical problem-solving visualizations. Traditional mathematics instruction relies heavily on step-by-step visual solutions written on blackboards, worksheets, and digital displays, where each symbol's placement, every equation's alignment, and the logical flow between steps conveys critical information that enables conceptual understanding (Arcavi, 2003; Presmeg, 2006). The ability to automatically generate such visualizations could revolutionize personalized tutoring (Heffernan & Heffernan, 2024) and improve accessibility for students with different learning styles (Boaler, 2016). Current T2I evaluation benchmarks, however, focus predominantly on natural image generation with metrics like FID and CLIP score (Heusel et al., 2017; Radford et al., 2021), artistic composition through DrawBench (Saharia et al., 2022) and PartiPrompts (Yu et al., 2022), or basic compositional understanding via T2I-CompBench++ (Huang et al., 2024), with mathematical content relegated to simple counting tasks ("generate 5 apples") or basic spatial relationships ("circle above square") (Huang et al., 2024; Kajić et al., 2024).

The recently introduced T2I-ReasonBench (Sun et al., 2025) includes mathematical reasoning as one of seven categories but allocates minimal attention to mathematical notation accuracy or multi-step procedural generation, while EduVisBench (Ji et al., 2025) touches on STEM visualization but evaluates only single-image generation rather than sequential mathematical procedures. Math2Visual (Wang et al., 2025) addresses elementary word problems by generating single illustrative images but does not tackle the complexity of high school mathematics with its demands for precise symbolic notation, multi-step solutions, and geometric constructions. This evaluation gap prevents us from understanding whether T2I models can serve as reliable tools for educational content generation, obscures the specific technical challenges that must be overcome to achieve mathematical accuracy, and leaves educators without standards for assessing AI-generated mathematical content (Alsayyad & Kadhem, 2025). The absence of comprehensive evaluation is concerning given that incorrect mathematical visualizations can reinforce misconceptions and impede learning (Vieriu & Petrea, 2025), making the development of comprehensive benchmarks not just a technical necessity but an educational imperative.

To address this gap, MathViz-Bench comprises 500 carefully curated problems sampled from levels 1-3 of the MATH dataset, spanning seven mathematical domains: Prealgebra, Algebra, Number Theory, Counting & Probability, Geometry, Intermediate Algebra, and Precalculus. Our benchmark provides an automated assessment pipeline consisting of three critical metrics: Sequential Consistency (SC) for evaluating logical flow across solution steps, Symbol Fidelity (SF) for assessing notation accuracy, and Mathematical Correctness (MC) for verifying calculation validity. The problems leverage the MATH dataset's three difficulty levels to systematically assess T2I model performance across increasing complexity, from Level 1 (basic single-step solutions) through Level 2 (intermediate problems) to Level 3 (complex multi-step reasoning), with each prompt requiring models to generate visual step-by-step solutions with proper mathematical notation and logical flow. The benchmark design prioritizes mathematical accuracy over aesthetic quality, recognizing that an educationally useful visualization must first and foremost convey correct mathematical information (Stylianides, 2007).

Our automated assessment pipeline leverages state-of-the-art Vision-Language Models (VLMs) such as Grok4 (xAI, 2025) to evaluate generated images across three metrics: **Sequential Consistency** assesses whether logical flow is maintained across solution steps by prompting VLMs to trace the mathematical reasoning from problem to solution and identify any breaks in logic or missing intermediate steps; **Symbol Fidelity** measures the accuracy of mathematical notation rendering by having VLMs identify and verify each mathematical symbol, checking for ambiguities between similar symbols (e.g., multiplication sign vs. letter 'x') and proper positioning of superscripts and subscripts; and **Mathematical Correctness** verifies the validity of mathematical operations and final answers by instructing VLMs to check each calculation, validate the mathematical approach, and confirm adherence to mathematical conventions. Each metric uses a structured 0-5 scoring rubric with detailed criteria, enabling consistent evaluation across different VLM judges (Zheng et al., 2024; Dubois et al., 2024). Each prompt includes structured metadata specifying required mathematical symbols, expected solution steps, critical spatial relationships, and common error patterns, enabling fine-grained analysis of model failures and providing clear targets for improvement. The use of vision-language models (VLMs) as evaluators enables scalable assessment across thousands of generated images and yields interpretable feedback about specific failure modes.

MathViz-Bench evaluation reveals fundamental architectural limitations in current T2I models. Our contributions are:

- **A comprehensive benchmark for mathematical visualization:** MathViz-Bench contains 500 problems from the MATH dataset (levels 1-3) across seven mathematical domains, each requiring T2I models to generate step-by-step visual solutions. We provide automated evaluation through three metrics scored 0-5: Sequential Consistency (logical flow), Symbol Fidelity (notation accuracy), and Mathematical Correctness (calculation validity).

- **Discovery of fundamental architectural limitations:** All models successfully render mathematical symbols (Symbol Fidelity: 1.81-4.57) but fail at mathematical reasoning (Mathematical Correctness: 0.65-4.05). Diffusion models show no improvement across difficulty levels or mathematical domains, proving they treat mathematics as visual patterns without understanding.

- **Identification of critical performance factors:** Models with integrated language understanding (GPT-Image-1: 84.05%, Gemini-2.5-Pro: 75.13%) outperform image-only diffusion models (FLUX1.1-Pro: 35.23%, WAN2.2: 28.05%, Stable Diffusion 3.5 Ultra: 22.05%) by 45-62 percentage points. Only GPT-Image-1 achieves educational viability with 62.3% high-quality outputs, establishing that mathematical visualization requires both symbolic reasoning and visual generation capabilities.

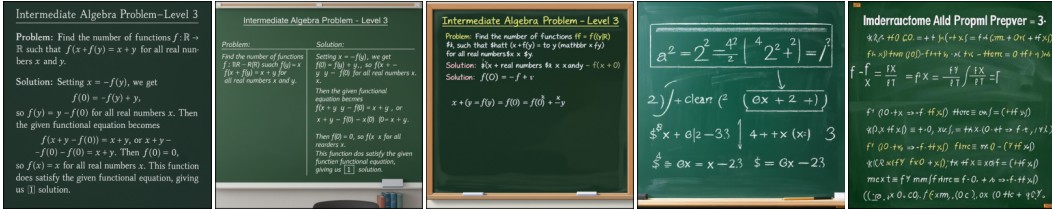

Figure 1: **Sample outputs from five T2I models.** From left to right: image outputs from GPT, GEMINI, FLUX, Stable Diffusion, and WAN. This example demonstrates the current capabilities and limitations of T2I models when generating mathematical content.

## 2 RELATED WORK

**Text-to-Image Generation Models.** Text-to-image generation has evolved from GANs (Reed et al., 2016; Zhang et al., 2017) to autoregressive models (Ramesh et al., 2021; 2022) and now diffusion models (Rombach et al., 2022; Saharia et al., 2022) that dominate the field. Recent advances include architectural innovations like Diffusion Transformers (Peebles & Xie, 2023), rectified flows (Liu et al., 2023; Esser et al., 2024), and various controllability mechanisms (Zhang et al., 2023; Mou et al., 2024; Ye et al., 2023). Despite these impressive capabilities for natural image generation, these models struggle with structured content requiring precise symbolic accuracy which is a critical limitation for mathematical visualization that our benchmark addresses.

**Text-to-Image Evaluation Benchmarks.** Existing T2I benchmarks focus primarily on natural image generation and basic compositional understanding. DrawBench (Saharia et al., 2022) and PartiPrompts (Yu et al., 2022) test challenging prompts, while T2I-CompBench++ (Huang et al., 2024) evaluates spatial relationships and attribute binding. VQA-based metrics (Hu et al., 2023; Lin et al., 2024) improve human correlation but don't address mathematical accuracy. Recent reasoning benchmarks touch on mathematical content superficially: T2I-ReasonBench (Wang et al., 2024) includes only basic counting tasks, while SPRIGHT (Lezama et al., 2024) and GenEval (Ghosh et al., 2024) evaluate spatial consistency and object counting without mathematical notation or multi-step procedures. This gap in evaluating mathematical symbol rendering and step-by-step solution generation motivates our work.

**AI Systems for Mathematical Tasks.** Prior work in AI for mathematics has explored various modalities. In the visual domain, benchmarks like MathVista (Lu et al., 2023a), MathVerse (Zhang et al., 2024), and Math2Visual (Wang et al., 2025) evaluate VQA models on mathematical reasoning from given images, with GPT-4V achieving only 49.9% on MathVista. Recent work on the

Gödel Test (Feldman & Karbasi, 2025b) evaluates GPT-5's ability to prove novel conjectures in combinatorial optimization, finding success with single-path reasoning but failure when combining insights across sources, mirroring our findings with visual generation. In the computational domain, SciML Agents (Gaonkar et al., 2025) evaluates LLMs' ability to generate executable code for solving ODEs, focusing on algorithmic correctness rather than visual representation. However, none of these approaches address the generation of accurate mathematical visualizations for educational purposes. MathViz-Bench fills this critical gap by evaluating T2I models' ability to create pedagogically meaningful visual content with correct notation, logical flow, and step-by-step solutions, thereby bridging the divide between computational solving and visual understanding.

## 3   THE MATHVIZ-BENCH DATASET

### 3.1   DATASET CONSTRUCTION

MathViz-Bench is constructed to systematically evaluate text-to-image models' capability to generate accurate mathematical visualizations for high school mathematics problems. We curate our dataset from the MATH dataset (Hendrycks et al., 2021), which contains mathematics problems across different difficult level with detailed step-by-step solutions. Our selection process ensures comprehensive coverage of mathematical domains.

### 3.2   PROBLEM SELECTION AND DISTRIBUTION

We sample 500 problems from MATH levels 1-3, corresponding to high school difficulty, excluding levels 4-5 which require advanced undergraduate knowledge. Our sampling strategy balances two objectives: ensuring equal representation across mathematical domains for fair comparison, and preserving the natural difficulty distribution within each domain as MATH dataset. Figure 2 illustrates the distribution of difficulty level at each mathematical domain.

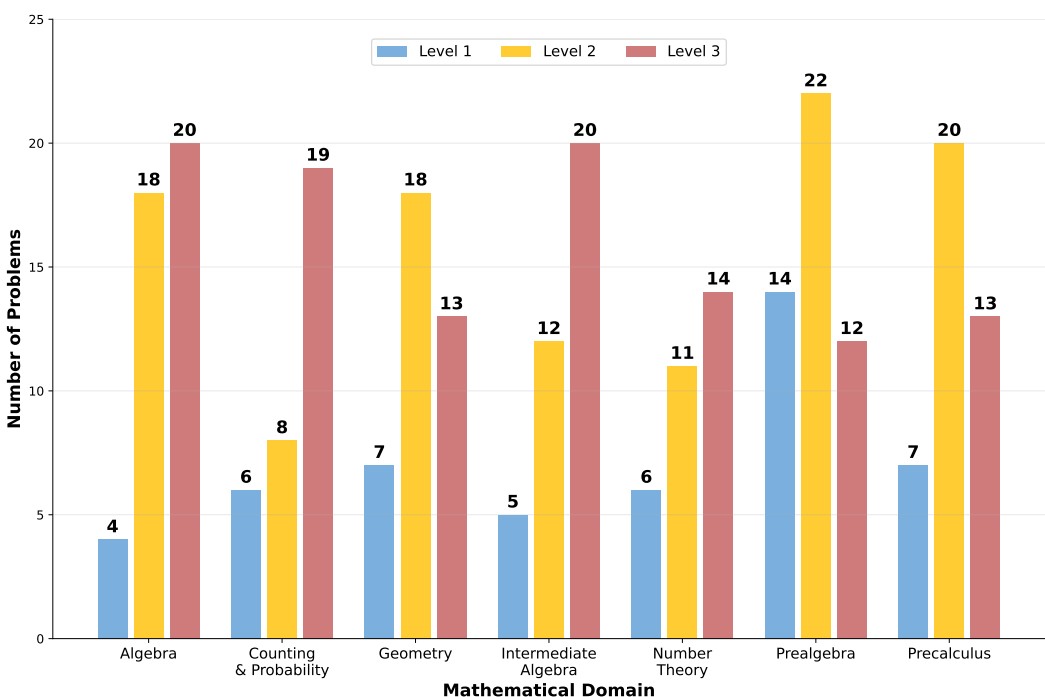

Figure 2: **Distribution of problems across mathematical domains by difficulty level in the extended MathViz-Bench dataset.** The dataset contains 500 problems with balanced representation across seven mathematical domains, ranging from 71-72 problems per domain. Each domain shows the distribution across three difficulty levels: Level 1 (blue), Level 2 (amber), and Level 3 (red).

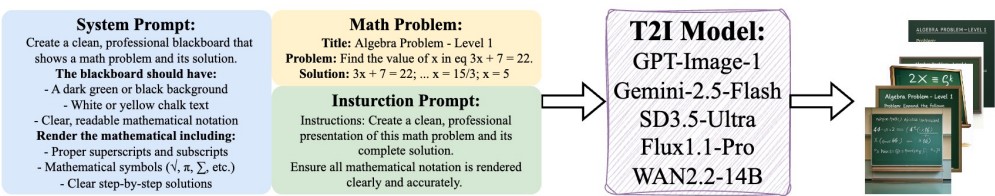

Figure 3: MathViz-Bench prompt pipeline. System prompts specify blackboard-style visual requirements (dark background, chalk text, clear mathematical notation), while instruction prompts combine MATH dataset problems with formatting directives. These prompts are fed to five T2I models to generate mathematical visualizations for evaluation.

## 3.3 Problem Characteristics

Each problem in MathViz-Bench contains three components:

**(1) Problem Statement:** Original MATH dataset question with LaTeX-formatted mathematical notation, ranging from basic arithmetic to advanced concepts including modular arithmetic, geometric constructions, and algebraic manipulations.

**(2) Complete Solution:** Step-by-step solution with intermediate calculations and final answer in `\boxed{}` format, serving as ground truth for Mathematical Correctness evaluation.

**(3) Metadata:** Domain classification (seven categories), difficulty level (1-3), enabling performance analysis across mathematical subfields and complexity gradients.

## 3.4 Prompt Formulation

Figure 3 illustrates our prompt formulation pipeline. We transform each MATH dataset problem into a structured prompt combining system instructions and problem content. System prompts specify blackboard-style visual requirements: dark background, white/yellow chalk text, and proper mathematical notation including superscripts, subscripts, fractions, and special symbols ($\pi$, $\sum$, $\sqrt{}$). Instruction prompts direct models to present problems with complete step-by-step solutions while ensuring accurate mathematical rendering. This dual-prompt structure tests models' ability to (1) correctly render complex mathematical notation and (2) maintain logical solution flow, simulating traditional mathematics instruction.

## 3.5 Comparison with Related Datasets

Table 1 compares MathViz-Bench with existing benchmarks that include mathematical content evaluation for visual generation or understanding. MathViz-Bench combines three critical requirements for evaluating mathematical content generation: dedicated focus on mathematics rather than general image generation, multi-step solution visualization requiring logical flow preservation, and precise symbolic notation rendering with proper mathematical formatting. While smaller than some benchmarks, our focused scope enables detailed error analysis and provides clear insights into specific failure modes of current T2I architectures when handling structured mathematical content.

Table 1: **Comparison of MathViz-Bench with related benchmarks.** We focus on mathematical visualization generation, unlike existing benchmarks that evaluate understanding or single-image illustration.

| Dataset | Task | Size | Math Focus | Multi-step | Symbolic |
|---|---|---|---|---|---|
| DrawBench | General T2I | 200 | ✗ | ✗ | ✗ |
| T2I-CompBench++ | Compositional T2I | 2,540 | ✗ | ✗ | ✗ |
| MathVista | VQA | 6,141 | ✓ | ✗ | Partial |
| Math2Visual | Math Word Problem | 1,200 | ✓ | ✗ | ✗ |
| **MathViz-Bench** | **Math Visualization** | **500** | ✓ | ✓ | ✓ |

# 4 EXPERIMENTS

We evaluate state-of-the-art commercial T2I models on MathViz-Bench to assess their capability in generating accurate mathematical visualizations. Our experiments are organized into four main analyses: (1) comprehensive comparison of overall model performance, (2) breakdown by mathematical domain to identify subject-specific strengths, (3) performance analysis across difficulty levels, and (4) systematic categorization of common failure modes.

## 4.1 EXPERIMENTAL SETUP

**Models.** We evaluate five leading commercial T2I models: Stable Diffusion 3.5 Ultra (SD) (Esser et al., 2024), Gemini-2.5-Pro (GEMINI) (Google, 2025), GPT-Image-1 (GPT) (OpenAI, 2025), FLUX1.1-Pro (FLUX) (Black Forest Labs, 2024), and WAN2.2 (WAN) (Wan et al., 2025). Each model is accessed through official APIs with default parameters, except for resolution which is standardized to 1024×1024 pixels to ensure fair comparison.

**Dataset.** We use our complete MathViz-Bench dataset comprising 500 problems sampled from levels 1-3 of the MATH dataset across seven mathematical domains: Prealgebra, Algebra, Number Theory, Counting & Probability, Geometry, Intermediate Algebra, and Precalculus. Each prompt is used to generate 1 images per model, resulting in 5 images per prompt across all models, totaling 1500 generated images for evaluation.

**Evaluation Protocol.** Each generated image is evaluated using our automated assessment pipeline with three metrics, each scored on a 0-5 scale:

- **Sequential Consistency (SC):** Evaluates logical flow across solution steps, checking whether each step follows logically from the previous, variables and notation are used consistently, and the visual layout maintains clear sequential progression.

- **Symbol Fidelity (SF):** Assesses the accuracy and clarity of mathematical symbol rendering, including distinction between similar symbols (e.g., x vs ×), proper positioning of superscripts/subscripts, and correct formation of fractions, roots, and grouping symbols.

- **Mathematical Correctness (MC):** Verifies the mathematical validity of the solution, including problem understanding, approach validity, calculation accuracy, and adherence to mathematical conventions. Critical errors (division by zero, invalid operations) cap this score at 2.

We employ Grok4 (xAI, 2025) as our primary evaluator using a structured prompt that produces JSON-formatted assessments. The evaluator provides detailed justification, identifies specific issues, and determines educational viability (all scores > 3). The total MathViz Score ranges from 0-15, with a recommendation classification: Good (12-15), Acceptable (9-11), or Bad (0-8).

## 4.2 COMPREHENSIVE MODEL COMPARISON

Table 2 reveals pronounced performance stratification among T2I models. GPT achieves 84.05% performance with 62.3% educational viability, while specialized diffusion models fail catastrophically below 40%. The universal pattern of highest Symbol Fidelity (1.81 to 4.57) versus lowest Mathematical Correctness (0.65 to 4.05) exposes a critical architectural limitation: current T2I models process mathematical symbols as visual tokens rather than semantic operators (Kajić et al., 2024). This decoupling explains why models can render perfect integral signs while producing mathematically nonsensical solutions, mirroring how neural networks can memorize patterns without learning underlying rules (Zhang et al., 2021).

The 45% performance gap between language-integrated and pure visual models reveals that mathematical visualization requires hierarchical reasoning absent in diffusion architectures (Lewkowycz et al., 2022). GPT and GEMINI leverage pre-trained mathematical knowledge from text corpora, enabling logical consistency checks impossible for models trained solely on image-caption pairs (Welleck et al., 2022). This advantage suggests that educational mathematical content generation necessitates multi-modal architectures that explicitly bridge symbolic and visual representations (Lu et al., 2023b).

Table 2: **Performance of commercial T2I models across three metrics.** GPT demonstrates clear superiority with 84% performance, while specialized diffusion models (FLUX, WAN, SD) fail across all evaluation criteria.

| Model | MC (0-5) | SC (0-5) | SF (0-5) | Total (0-15) | Overall (%) | High Quality (Score>12) (%) |
|---|---|---|---|---|---|---|
| GPT | 4.05 | 3.98 | 4.57 | 12.60 | 84.05% | 62.3% |
| GEMINI | 3.49 | 3.49 | 4.29 | 11.27 | 75.13% | 41.2% |
| FLUX | 1.25 | 1.44 | 2.60 | 5.29 | 35.23% | 4.8% |
| WAN | 0.90 | 1.17 | 2.14 | 4.21 | 28.05% | 0.8% |
| SD | 0.65 | 0.85 | 1.81 | 3.31 | 22.05% | 0.2% |

## 4.3 Performance Across Mathematical Domains

Figure 4 reveals stark architectural differences across mathematical domains. Language-integrated models maintain consistent performance: GPT-Image-1 (3.8-4.5) peaks in Intermediate Algebra (4.5) and Geometry (4.4), while Gemini-2.5-Pro achieves 3.3-3.9 across domains. No systematic advantage emerges for discrete versus continuous mathematics, contradicting expectations that discrete problems would better suit transformer architectures (Polu & Sutskever, 2020). Diffusion models demonstrate complete domain invariance with scores below 2.2: FLUX1.1-Pro (1.0-1.9), Stable Diffusion 3.5 Ultra (0.9-1.2), and WAN2.2 (1.0-2.2) show standard deviations under 0.3, confirming absence of mathematical reasoning. Even Geometry, where spatial reasoning might theoretically advantage visual models, yields no improvement (FLUX: 1.8, SD: 1.0, WAN: 1.3). This uniform failure indicates these models perform surface pattern matching without semantic understanding (Zhang et al., 2024), establishing that mathematical visualization requires logical inference beyond visual processing (Seo et al., 2015).

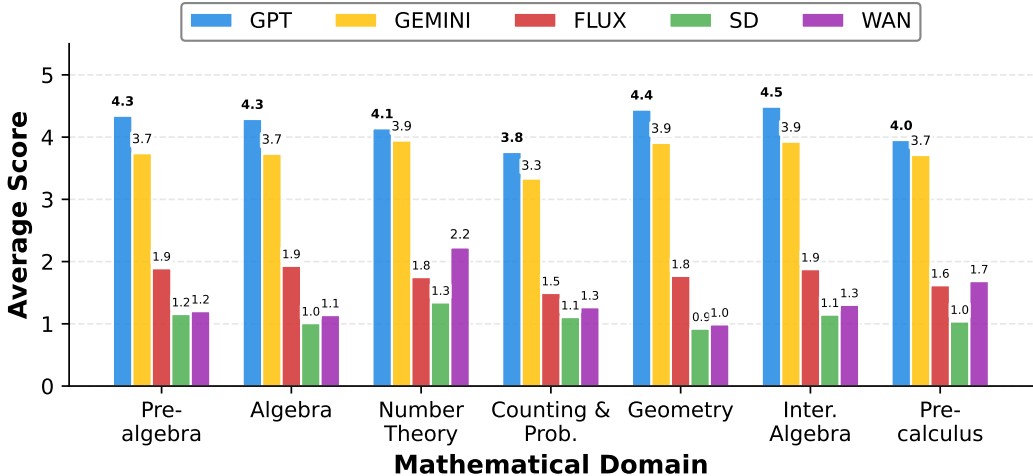

Figure 4: **Performance across mathematical domains for five T2I models.** GPT maintains consistently high scores (3.8-4.5) with strongest performance in Intermediate Algebra (4.5) and Geometry (4.4). GEMINI shows moderate capability (3.3-3.9) across domains. Pure diffusion models (FLUX, SD, WAN) fails with scores below 2.2, demonstrating domain-invariant failure.

## 4.4 Performance Across Difficulty Levels

Figure 5 reveals distinct architectural failure patterns. GPT shows gradual decline from 4.33 at Level 1 to 4.10 at Level 2 and 3.84 at Level 3, maintaining competence despite increasing complexity. GEMINI drops sharply from 3.71 to 3.40 between Levels 1-2 then stabilizes at 3.37, suggesting error propagation primarily impacts the transition to multi-step problems (Lightman et al., 2023).

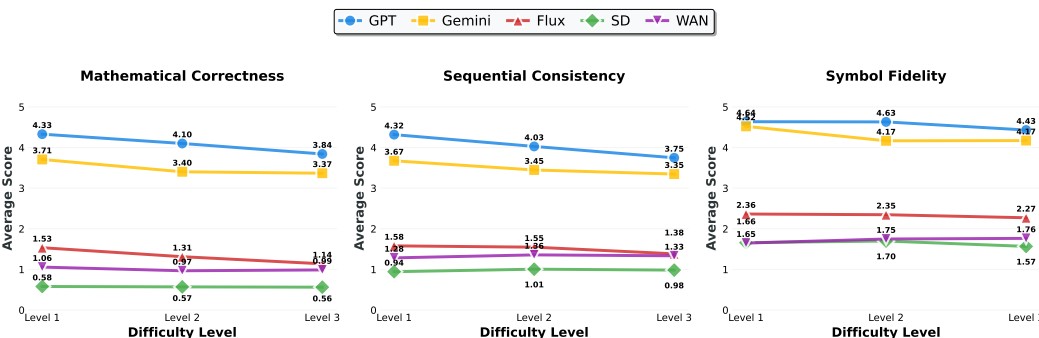

Figure 5: **Performance progression across difficulty levels.** GPT shows declining performance with complexity, dropping from 4.33 (Level 1) to 3.84 (Level 3) in Mathematical Correctness, while maintaining stable Symbol Fidelity (4.43-4.64). GEMINI similarly declines from 3.71 to 3.37 in Mathematical Correctness. Diffusion models (FLUX, WAN, SD) demonstrate complete difficulty invariance with Mathematical Correctness remaining below 1.53 across all levels. Symbol Fidelity remains the strongest metric across all models, highlighting the disconnect between visual rendering and mathematical understanding.

Diffusion models exhibit complete difficulty invariance: FLUX declines marginally from 1.53 to 1.14, SD remains flat at approximately 0.57, and WAN hovers around 1.0 across all levels, with total variations under 0.4 points. This flat performance at near-zero Mathematical Correctness, contrasted with maintained Symbol Fidelity (all models above 1.57), confirms these architectures lack mathematical reasoning capabilities entirely rather than being constrained by problem complexity (Feldman & Karbasi, 2025a). The binary nature of mathematical competence indicates that incremental improvements to diffusion architectures cannot achieve educational viability without fundamental incorporation of symbolic reasoning (Power et al., 2022).

## 4.5 ERROR PATTERN ANALYSIS AND FAILURE MODES

Analysis of 2,500 generated images reveals systematic failure patterns illuminating fundamental limitations in T2I mathematical content generation.

Table 3: **Error type distribution across model architectures.** Multimodal models (GPT, GEM-INI) maintain balanced error distributions with only 1.1% critical failures, while diffusion models concentrate 41.7% of errors in symbol rendering with 33.8% critical failure rates.

| Error Type | Multimodal (GPT, GEMINI) | Diffusion (FLUX, SD, WAN) | Overall |
|---|---|---|---|
| Symbol Rendering | 18.4% | 41.7% | 33.2% |
| Sequential Logic | 24.1% | 35.8% | 32.1% |
| Math Correctness | 22.3% | 31.2% | 28.4% |
| Format/Layout | 4.8% | 8.1% | 6.3% |
| **Mean Score** | 11.80/15 | 4.10/15 | – |
| **Critical Failures** | 1.1% | 33.8% | 20.6% |

Table 3 reveals architectural stratification: multimodal models maintain balanced error distributions with low critical failure rates (1.1%), while diffusion models concentrate failures in symbol rendering (41.7%) with failure rates (33.8%). This 7.7% performance gap confirms that mathematical visualization requires language-vision integration (Zhang et al., 2024; Lu et al., 2023a).

Table 4 quantifies symbol-specific failures. Complex notation (fractions, summations) fail at 20-35% rates, while basic operators show lower but still problematic rates. The 34.8% failure rate for summation symbols indicates current models cannot handle multi-level mathematical notation (Bosheah & Bilicki, 2025).

Table 4: **Critical symbol failure analysis for mathematical notation.** Complex symbols (fractions, summations) fail at 20-35% rates while basic operators show lower but educationally problematic rates, with summation symbols achieving 34.8% failure rate indicating inability to handle multi-level notation.

| Symbol Type | Instances | Failures | Failure Rate | Impact |
|---|---|---|---|---|
| Fractions ($\frac{a}{b}$) | 423 | 87 | 20.6% | High |
| Exponents ($x^n$) | 312 | 52 | 16.7% | High |
| Summation ($\sum$) | 89 | 31 | 34.8% | Medium |
| Equality (=) | 847 | 31 | 3.7% | Critical |
| Parentheses | 621 | 18 | 2.9% | Low |

**Insights.** The 32% performance difference in critical failure rates between multimodal (1.1%) and diffusion models (33.8%) establishes that mathematical visualization requires semantic understanding, not visual pattern matching. Diffusion models concentrate 41.7% of errors in symbol rendering, failing at character formation rather than layout. The 34.8% summation symbol failure rate reveals inability to process hierarchical notation, treating $\sum$ as an atomic visual unit rather than understanding its compositional structure. Even basic equality signs fail at 3.7%, indicating models cannot reliably distinguish visually similar characters. Multimodal models distribute errors evenly across symbol rendering (18.4%), sequential logic (24.1%), and mathematical correctness (22.3%), demonstrating balanced capability across all aspects. In contrast, diffusion models' concentration in rendering failures (41.7%) reveals they lack the fundamental integration of symbol recognition, logical sequencing, and semantic validation that mathematical visualization demands. These capabilities emerge only through language-vision architectures that can simultaneously process visual form and mathematical meaning.

## 4.6 COMPUTATIONAL EFFICIENCY ANALYSIS

Analysis of 2,500 generations reveals a critical efficiency paradox for mathematical T2I deployment.

**Generation Timing.** Mean generation times cluster tightly: WAN (11.33s), FLUX (12.24s), SD (13.13s), GPT (14.17s), GEMINI (14.36s). The 3.03s range (27% variation) contrasts sharply with the 14-fold quality gap in mathematical correctness. Low standard deviations (3.39-5.67s) indicate consistent performance across architectures.

**Quality-Efficiency Trade-off.** Despite 25% slower generation, GPT delivers 19.27 quality points per minute versus WAN's 5.68, a 3.4x efficiency advantage when quality-adjusted. All models exceed educational capacity requirements (>2,000 visualizations per 8-hour period), making speed differences negligible for practical deployment. The 20% time premium for GPT yields 400% quality improvement, demonstrating that mathematical visualization requires quality-centric rather than latency-centric optimization (Heffernan & Heffernan, 2024).

## 5 CONCLUSION

MathViz-Bench provides the first systematic evaluation of T2I models' mathematical visualization capabilities, revealing a fundamental architectural limitation: current models cannot bridge visual generation with mathematical reasoning. The 45-62 percentage point performance gap between language-integrated and diffusion architectures, combined with universal decoupling of Symbol Fidelity from Mathematical Correctness, demonstrates that mathematical content generation requires semantic understanding absent in pure visual models. Diffusion models' complete difficulty and domain invariance (variance<0.08) indicates binary capability where models either possess mathematical reasoning or lack it entirely. The 32% difference in critical failure rates (1.1% vs 33.8%) confirms that language grounding is necessary but insufficient, as even top-performing models achieve only 62.3% educational viability. The 34.8% failure rate for hierarchical notation reveals fundamental limitations in compositional understanding that cannot be addressed through scale alone. MathViz-Bench provides both the evaluation framework and performance baselines necessary for developing educationally viable mathematical visualization systems.

ETHIC STATEMENT

This paper does not involve human subjects, personally identifiable data, or sensitive applications. We do not foresee direct ethical risks. We follow the ICLR Code of Ethics and affirm that all aspects of this research comply with the principles of fairness, transparency, and integrity.

REPRODUCIBILITY STATEMENT

We ensure reproducibility on both theoretical and empirical fronts. For theory, we include all formal assumptions, definitions, and complete proofs in the appendix. For experiments, we describe model architectures, datasets, preprocessing steps, hyperparameters, and training details in the main text and appendix. Code and scripts are provided in the supplementary materials to replicate the empirical results.

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

# Appendix

## A   ILLUSTRATION OF T2I MODELS OUTPUTS

We shows the outputs of the five T2I models using the math problem:

**Problem:** [Let $f(x) = 2x^4 + x^3 + x^2 - 3x + r$. For what value of $r$ is $f(2) = 0$?]

Algebra Problem – Level 2

Problem: Let $f(x) = 2x^4 + x^3 + x^2 - 3x + r$.
For what value of $r$ is $f(2) = 0$?

Solution: Evaluating gives
$$f(2) = 2(2)^4 + (2)^3 + (2)2^2 - 3(2) + r$$
$$= 32 + 8 + 4 - 6 + r =$$
$$= 38 + r.$$
This is equal to 0 when $r = -38$.
$$r = \boxed{-38.}$$

Figure 6: **Output From GPT-Image-1**

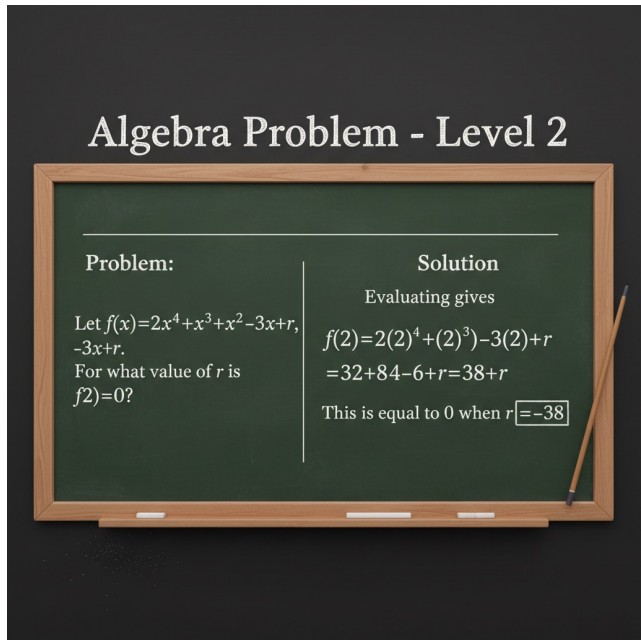

Figure 7: **Output From Gemini-2.5-Pro**

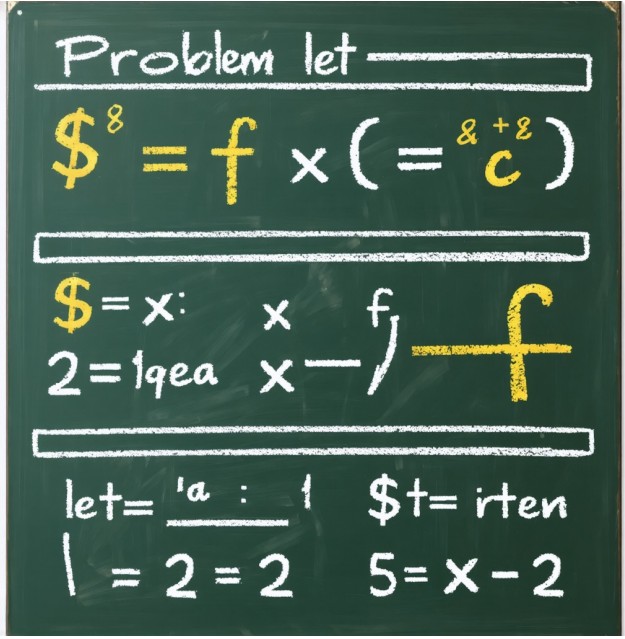

Figure 8: **Output From Flux1.1-Pro**

Figure 9: **Output From Stable Diffusion 3.5 Ultra**

Figure 10: **Output From WAN2.2**

## B  COST ANALYSIS

This section provides a comprehensive cost analysis for generating the 500 mathematical visualizations used in MathViz-Bench across five commercial T2I models. The cost data reflects actual expenditures during the benchmark evaluation and provides insights for researchers considering similar large-scale evaluations.

### B.1  GENERATION COSTS BY MODEL

Table 5 summarizes the total costs incurred for generating 500 mathematical visualization images across all evaluated models. Costs represent actual API charges during the evaluation period.

Table 5: Cost analysis for generating 500 mathematical visualizations across commercial T2I models.

| Model | Total Cost (USD) | Cost per Image (USD) | Performance (0-15) | Cost per Quality Point (USD) |
|-------|------------------|----------------------|--------------------|------------------------------|
| GPT-Image-1 | $100 | $0.200 | 12.60 | $0.016 |
| Stable Diffusion | $40 | $0.080 | 3.31 | $0.024 |
| Gemini-2.5-Pro | $25 | $0.050 | 11.27 | $0.004 |
| FLUX1.1-Pro | $20 | $0.040 | 5.29 | $0.008 |
| WAN2.2 | $20 | $0.040 | 4.21 | $0.010 |
| *Total* | $205 | $0.082 | 7.34 | $0.012 |

### B.2  COST-PERFORMANCE ANALYSIS

The cost analysis reveals significant disparities in value proposition across models. GEMINI emerges as the most cost-effective option, achieving 11.27/15 performance at only $0.004 per quality point which is four times more efficient than the next best model. Despite being the most expensive per image ($0.200), GPT demonstrates reasonable cost efficiency ($0.016 per quality point) due to its superior performance.

In contrast, specialized diffusion models show poor cost-performance ratios. Stable Diffusion, despite moderate per-image costs ($0.080), delivers the worst cost efficiency ($0.024 per quality point) due to catastrophically low performance (3.31/15). FLUX and WAN, while offering the lowest absolute costs ($0.040 per image), provide limited value given their poor mathematical visualization capabilities.

### B.3  BUDGET RECOMMENDATIONS

For researchers planning similar evaluations, our cost analysis suggests three deployment strategies:

**High-Quality Research:** Allocate $100-200 per 500 images for GPT-class models when mathematical accuracy is paramount. The 20¢ per image premium delivers educational-grade quality essential for rigorous evaluation.

**Balanced Evaluation:** GEMINI provides optimal cost-performance balance at $25 per 500 images (5¢ per image), achieving 75% of GPT's quality at 25% of the cost. Suitable for large-scale studies with budget constraints.

**Baseline Comparison:** Budget $20-40 per 500 images for specialized diffusion models when establishing performance baselines. While these models fail for practical applications, they provide valuable negative controls for research evaluation.

## LLM USAGE DISCLOSURE

LLMs were used only to polish language, such as grammar and wording. These models did not contribute to idea creation or writing, and the authors take full responsibility for this paper's content.

