# OpenReview forum: "MathViz-Bench: Evaluating Text-to-Image Models on Visually Solving Math Problems"
_ICLR.cc/2026/Conference — ICLR 2026 Conference Desk Rejected Submission_

### Official Review · Reviewer_ytbB · 2025-10-31

**Soundness:** 3
**Presentation:** 3
**Contribution:** 2
**Rating:** 2
**Confidence:** 4

**Summary:**

This paper proposes a new benchmark, named MathViz-Bench, to assess the abilities of text-to-image (T2I) models for accurate and consistent rendering step-by-step solutions of high school mathematics problems. To do this, the authors sample 500 problems from existing MATH dataset, and then utilize the existing five T2I models to generate the visualizations. Finally, they use Grok4 as a judge to assess the qualities of generated results. In experimental result, it was shown that multimodal models (GPT, GEMINI) perform better than diffusion models (FLUX, SD, WAN).

**Strengths:**

- This paper tackles the missing gap of the current benchmarks and studies the interesting task of visualizing math solutions.
- The paper presents comprehensive analysis of evaluation comparisons with five T2I models.
- Writing is easy to read and good, in general.

**Weaknesses:**

- Even though it is interesting, the significance of the proposed benchmark is questionable. Visualizing math solutions may not be necessary as it is possible to directly rendering them by using LaTeX. This may be easier to understand and generate more accurate visualizations.
- This paper generates benchmark by using T2I models. But there is no human annotation/evaluation for it, so the value of this created benchmark is also questionable.
- Furthermore, there is no human evaluation for TI2 model outputs, which limits the effectiveness of the presented analysis.
- In addition, the findings from the experiment show no surprise. It is expected that multimodal models perform better than diffusion models and no further analysis or suggestion to improve was presented.
- Finally, this paper lacks to present a new T2I model to improve the current T2I models. It is expected, with the newly proposed benchmark, to present a new improved T2I model perform better on the new benchmark.

**Questions:**

Please see and address the weaknesses above.

- Typo
#042: bosheah2025challenges -> typo?

---

### Official Review · Reviewer_KCE7 · 2025-11-01

**Soundness:** 2
**Presentation:** 3
**Contribution:** 2
**Rating:** 2
**Confidence:** 4

**Summary:**

This paper presents MathViz-Bench, a benchmark for evaluating text-to-image models on sequential mathematical visualization. The benchmark contains 500 high-school problems from MATH levels 1 to 3 and uses three metrics to assess sequential consistency, symbol fidelity, and mathematical correctness via an automated VLM judge. Experiments demonstrate that language-integrated models significantly outperform diffusion-only approaches, exposing a substantial gap between visual polish and symbolic precision while offering valuable guidance for improving model design.

**Strengths:**

1. This paper raises an important and underexplored problem: evaluating text-to-image models on step-by-step mathematical visualization.

2. The paper is well-written, with clear figures and detailed explanations that make the work easy to follow.

3. The evaluation framework is thoughtfully designed, with diagnostic metrics that disentangle presentation quality from mathematical validity.

4. The study provides well-controlled comparisons of leading multimodal and diffusion models. This is complemented by a fine-grained error analysis and suggests actionable research directions.

**Weaknesses:**

1. The dataset scope is limited to high-school difficulty, and its overall size is relatively small. Because the benchmark includes only MATH levels 1–3, the results may not generalize to more advanced mathematics. In addition, MathViz-Bench is smaller than several widely used T2I or VQA benchmarks, which may potentially be less able to show true gaps and less comprehensive in scope.

2. The evaluation employs a fixed blackboard presentation style without testing alternative layouts such as paper worksheets, whiteboards, slides, or textbook formatting, which may introduce style-induced bias.

3. The evaluation relies on a VLM to assign scores without human validation, potentially introducing bias and errors that could affect model rankings and conclusions.

**Questions:**

1. The evaluation is limited to a single image generation per problem for each model. I strongly encourage the authors to conduct multiple sampling runs and systematically investigate how randomness and various decoding strategies affect performance.

2. The current evaluation is quite limited in scope. It would be stronger to include a broader set of T2I systems, especially more open-source and fine-tuned models, to improve external validity and potentially reveal more informative findings.

---

### Official Review · Reviewer_Pj9e · 2025-11-03

**Soundness:** 2
**Presentation:** 2
**Contribution:** 1
**Rating:** 2
**Confidence:** 3

**Summary:**

The paper introduces MathViz-Bench, a benchmark sampled from the MATH dataset that evaluates text-to-image models on step-by-step mathematical visualization.

**Strengths:**

This paper provides empirical analysis that reveals the architectural limitations in the task of step-by-step mathematical visualization.

**Weaknesses:**

1. The motivation of this dataset is confusing. It is built on (i.e., sampling) the existing dataset MATH. Moreover, based on the description in Section 3, it looks like the inputs to VLMs include the math problem, complete step-by-step solution, and additional style constraints. Is this dataset only testing models' capability for solution visualization without solution generation?

2. The studied dataset has limited scope with only 500 problems restricted to "blackboard-style" solution visualization. Moreover, the math problems are limited to algebraic questions without the critical mathematical visualization types such as geometric diagrams, graphs, plots, and visual proofs.

3. The presentation is unclear and important details are missing. For example, Figure 1 is intended to illustrate the limitation of T2I models, but it didn't discuss what limitations are and why it is challenging. Moreover, the evaluation details are missing in Section 4: what does score 0-5 mean? Why are "educational viability" defined by "all scores > 3"? Is this based on pedagogical justification or empirical validation?

4. The evaluation is based on LLM judge (Grok4), but it is unclear why it is selected as the judge model and how reliable the evaluation method is. Human evaluation is needed and it is important to report the agreement scores with human judge.

5. The evaluation experiment is limited with only commercial T2I models tested.

6. The authors attribute performance differences to architectural factors (language integration vs. pure visual) without controlling other factors such as model size the same.

**Questions:**

1. One typo in Line 42.

---

### Note · Program_Chairs · 2026-01-17
**Submission Desk Rejected by Program Chairs**

The following references in this submission do not refer to real documents and/or have major errors in bibliographic information:

 Pan Lu et al. Unified multi-modal learning for vision, language and mathematics. ICLR, 2023b
Neil T Heffernan and Cristina L Heffernan. Intelligent tutoring systems in mathematics education: A 2024 review. International Journal of Artificial Intelligence in Education, 34:123-145, 2024